# Local chromosome context is a major determinant of crossover pathway biochemistry during budding yeast meiosis

Darpan Medhi[1,2,3], Alastair SH Goldman[2,3], Michael Lichten[1]*

[1]Laboratory of Biochemistry and Molecular Biology, Center for Cancer Research, National Cancer Institute, Bethesda, United States; [2]Sheffield Institute for Nucleic Acids, The University of Sheffield, Sheffield, United Kingdom; [3]Department of Molecular Biology and Biotechnology, The University of Sheffield, Sheffield, United Kingdom

**Abstract** The budding yeast genome contains regions where meiotic recombination initiates more frequently than in others. This pattern parallels enrichment for the meiotic chromosome axis proteins Hop1 and Red1. These proteins are important for Spo11-catalyzed double strand break formation; their contribution to crossover recombination remains undefined. Using the sequence-specific *VMA1*-derived endonuclease (VDE) to initiate recombination in meiosis, we show that chromosome structure influences the choice of proteins that resolve recombination intermediates to form crossovers. At a Hop1-enriched locus, most VDE-initiated crossovers, like most Spo11-initiated crossovers, required the meiosis-specific MutLγ resolvase. In contrast, at a locus with lower Hop1 occupancy, most VDE-initiated crossovers were MutLγ-independent. In *pch2* mutants, the two loci displayed similar Hop1 occupancy levels, and VDE-induced crossovers were similarly MutLγ-dependent. We suggest that meiotic and mitotic recombination pathways coexist within meiotic cells, and that features of meiotic chromosome structure determine whether one or the other predominates in different regions.

*For correspondence: mlichten@
helix.nih.gov

**Competing interests:** The authors declare that no competing interests exist.

## Introduction

The transition from the mitotic cell cycle to meiosis involves substantial changes in mechanisms of DNA double strand break (DSB) repair by homologous recombination (HR). Most mitotic HR repairs spontaneous lesions, and most repair products are non-crossovers (NCOs) that do not involve exchange of flanking parental sequences (*Kadyk and Hartwell, 1992*; *Ira et al., 2003*; *Pâques et al., 1998*). In contrast, meiotic recombination is initiated by programmed DSBs (*Cao et al., 1990*; *Sun et al., 1989*) that often are repaired as crossovers (COs) between homologous chromosomes (homologs), with exchange of flanking parental sequences. Inter-homolog COs, combined with sister chromatid cohesion, create physical linkages that ensure faithful homolog segregation during the first meiotic division, avoiding chromosome nondisjunction and consequent aneuploidy in gametes (reviewed by *Hunter, 2015*).

The DSBs that initiate meiotic recombination are formed by Spo11 in complex with a number of accessory proteins, and will be referred to here as Spo11-DSBs (reviewed by *Lam and Keeney, 2015*). Spo11-DSBs and resulting recombination events are non-uniformly distributed in the genomes of organisms ranging from budding yeast to humans (*Baudat and Nicolas, 1997*; *Blitzblau et al., 2007*; *Buhler et al., 2007*; *Fowler et al., 2013*; *Gerton et al., 2000*; *Hellsten et al.,*

**eLife digest** Inside the cells of many species, double-stranded DNA is packaged together with specialized proteins to form structures called chromosomes. Breaks that span across both strands of the DNA can cause cell death because if the break is incorrectly repaired, a segment of the DNA may be lost. Cells use a process known as homologous recombination to repair such breaks correctly. This uses an undamaged DNA molecule as a template that can be copied to replace missing segments of the DNA sequence. During the repair of double-strand breaks, connections called crossovers may form. This results in the damaged and undamaged DNA molecules swapping a portion of their sequences.

In meiosis, a type of cell division that produces sperm and eggs, cells deliberately break their chromosomes and then repair them using homologous recombination. The crossovers that form during this process are important for sharing chromosomes between the newly forming cells. It is crucial that the crossovers form at the right time and place along the chromosomes.

Chromosomes have different structures depending on whether a cell is undergoing meiosis or normal (mitotic) cell division. This structure may influence how and where crossovers form. Enzymes called resolvases catalyze the reactions that occur during the last step in homologous recombination to generate crossovers. One particular resolvase acts only during meiosis, whereas others are active in both mitotic and meiotic cells. However, it is not known whether local features of the chromosome structure – such as the proteins packaged in the chromosome alongside the DNA – influence when and where meiotic crossover occurs.

Medhi et al. have now studied how recombination occurs along different regions of the chromosomes in budding yeast cells, which undergo meiosis in a similar way to human cells. The results of the experiments reveal that the mechanism by which crossovers form depends on proteins called axis proteins, one type of which is specifically found in meiotic chromosomes. In regions that had high levels of meiotic axis proteins, crossovers mainly formed using the meiosis-specific resolvase enzyme. In regions that had low levels of meiotic axis proteins, crossovers formed using resolvases that are active in mitotic cells. Further experiments demonstrated that altering the levels of one of the meiotic axis proteins changed which resolvase was used.

Overall, the results presented by Medhi et al. show that differences in chromosome structure, in particular the relative concentration of meiotic axis proteins, influence how crossovers form in yeast. Future studies will investigate whether this is observed in other organisms such as humans, and whether local chromosome structure influences other steps of homologous recombination in meiosis.

2013; *Pratto et al., 2014*; *Singhal et al., 2015*; *Smagulova et al., 2011*; *Wijnker et al., 2013*). In budding yeast, this non-uniform distribution of Spo11-DSBs is influenced by meiosis-specific proteins, Red1 and Hop1, which are components of the meiotic chromosome axis. The meiotic chromosome axis coordinates sister chromatids and forms the axial element of the synaptonemal complex, which holds homologs in tight juxtaposition (*Hollingsworth et al., 1990*; *Page and Hawley, 2004*; *Smith and Roeder, 1997*). Spo11-DSBs form frequently in large (ca 50–200 kb) 'hot' domains that are also enriched for Red1 and Hop1, and these 'hot' domains are interspersed with similarly-sized 'cold' regions where Spo11-DSBs are infrequent and Red1/Hop1 occupancy levels are low (*Baudat and Nicolas, 1997*; *Blat et al., 2002*; *Blitzblau et al., 2007*; *Buhler et al., 2007*; *Panizza et al., 2011*). Normal Spo11-DSB formation requires recruitment of Spo11 and accessory proteins to the meiotic axis (*Panizza et al., 2011*; *Prieler et al., 2005*), and Red1/Hop1 are also central to mechanisms that direct Spo11-DSB repair towards use of the homolog as a recombination partner (*Carballo et al., 2008*; *Niu et al., 2005*; *Schwacha and Kleckner, 1997*). Other eukaryotes contain Hop1 analogs that share a domain, called the HORMA domain (*Rosenberg and Corbett, 2015*), and correlations between these meiotic axis proteins and DSB formation are observed in fission yeast, nematodes and in mammals (*Fowler et al., 2013*; *Goodyer et al., 2008*; *Wojtasz et al., 2009*). Thus, most meiotic interhomolog recombination occurs in the context of a specialized chromosome structure and requires components of that structure.

Meiotic recombination pathways diverge after DSB formation and homolog-directed strand invasion. In budding yeast, about half of meiotic events form NCOs via synthesis-dependent strand annealing, a mechanism that does not involve stable recombination intermediates (*Allers and Lichten, 2001a*; *McMahill et al., 2007*) and is suggested to be the predominant HR pathway in mitotic cells (*Bzymek et al., 2010*; *McGill et al., 1989*). Most of the remaining events are repaired by a meiosis-specific CO pathway, in which an ensemble of meiotic proteins, called the ZMM proteins, stabilize early recombination intermediates and promote their maturation into double Holliday junction joint molecules (*Allers and Lichten, 2001a*; *Börner et al., 2004*; *Lynn et al., 2007*; *Schwacha and Kleckner, 1994*). These ZMM-stabilized joint molecules (JMs) are subsequently resolved as COs (*Sourirajan and Lichten, 2008*) through the action of the MutLγ complex, which contains the Mlh1, Mlh3, and Exo1 proteins (*Argueso et al., 2004*; *Khazanehdari and Borts, 2000*; *Wang et al., 1999*; *Zakharyevich et al., 2010*, *2012*). MutLγ does not appear to make significant contributions to mitotic COs (*Ira et al., 2003*). A minority of events form ZMM-independent JMs that are resolved as both COs and NCOs by the structure-selective nucleases (SSNs) Mus81-Mms4, Yen1, and Slx1-Slx4, which are responsible for most JM resolution during mitosis (*Argueso et al., 2004*; *Santos et al., 2003*; *De Muyt et al., 2012*; *Ho et al., 2010*; *Muñoz-Galván et al., 2012*; *Zakharyevich et al., 2012*; reviewed by *Wyatt and West, 2014*). A similar picture, with MutLγ forming most meiotic COs and SSNs playing a minor role, is observed in several other eukaryotes (*Berchowitz et al., 2007*; *Holloway et al., 2008*; *Plug et al., 1998*).

To better understand the factors that promote the unique biochemistry of CO formation during meiosis, in particular MutLγ-dependent JM resolution, we considered two different hypotheses. In the first, expression of meiosis-specific proteins and the presence of high levels of Spo11-DSBs results in nucleus-wide changes in recombination biochemistry, shifting its balance towards MutLγ-dependent resolution of JMs, wherever they might occur. In the second, local features of meiotic chromosome structure, in particular enrichment for meiosis-specific chromosome axis proteins, provides an in cis structural environment that favors MutLγ-dependent JM resolution. However, because Spo11-DSBs form preferentially in Red1/Hop1-enriched regions, and because these proteins are required for efficient Spo11-DSB formation and interhomolog repair, it is difficult to distinguish these two models by examining Spo11-initiated recombination alone.

To test these two hypotheses, we developed a system in which meiotic recombination is initiated by the sequence- and meiosis-specific *VMA1* derived endonuclease, VDE (*Gimble and Thorner, 1992*; *Nagai et al., 2003*). VDE initiates meiotic recombination at similar levels wherever its recognition sequence (*VRS*) is inserted (*Fukuda et al., 2008*; *Neale et al., 2002*; *Nogami et al., 2002*). VDE-catalyzed DSBs (hereafter called VDE-DSBs) form independent of Spo11 and meiotic axis proteins. However, like Spo11-DSBs, VDE-DSBs form after pre-meiotic DNA replication and are repaired using end-processing and strand invasion activities that also repair Spo11-DSBs (*Fukuda et al., 2003*; *Neale et al., 2002*). We examined resolvase contributions to VDE-initiated CO formation, and obtained evidence that local enrichment for meiotic axis proteins promotes MutLγ-dependent CO formation; while recombination that occurs outside of this specialized environment forms COs by MutLγ-independent mechanisms. We also show that CO formation at a locus, and in particular MutLγ-dependent CO formation, requires Spo11-DSB formation elsewhere in the genome.

## Results

### Using VDE to study meiotic recombination at 'hot' and 'cold' loci

The recombination reporter used for this study contains a VDE recognition sequence (*VRS*) inserted into a copy of the *ARG4* gene on one chromosome, and an uncleavable mutant recognition sequence (*VRS103*) on the homolog (*Figure 1*). Restriction site polymorphisms at flanking *Hin*dIII sites, combined with the heterozygous *VRS* site, allow differentiation of parental and recombinant DNA molecules. This recombination reporter was inserted at two loci: *HIS4* and *URA3*, which are 'hot' and 'cold', respectively, for Spo11-initiated recombination and Red1/Hop1 occupancy (*Borde et al., 1999*; *Buhler et al., 2007*; *Panizza et al., 2011*; *Wu and Lichten, 1995*; also see Figure 4A and Figure 4—figure supplement 1, below). Consistent with previous reports, Spo11-DSBs and the resulting crossovers, are about five times more frequent in inserts at *HIS4* than at *URA3* (*Figure 1—figure supplement 1A*). When VDE is expressed, ~90% of *VRS* sites at both loci were

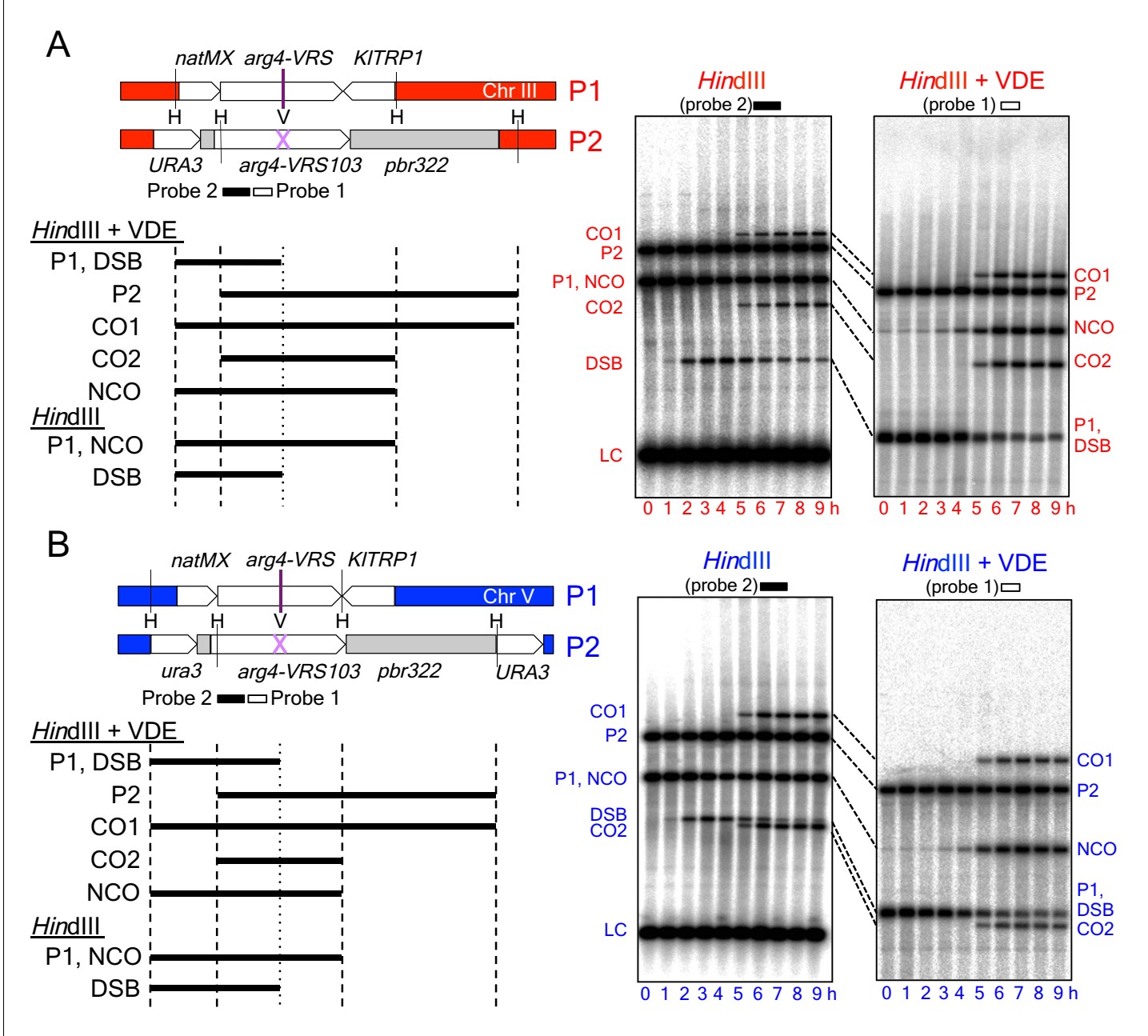

**Figure 1.** Inserts used to monitor VDE-initiated meiotic recombination. The *HIS4* and *URA3* loci are denoted throughout this paper in red and blue, respectively, and are in Red1/Hop1 enriched and depleted regions, respectively (see *Figure 4A* and *Figure 4—figure supplement 1*, below). (A) Left— map of VDE-reporter inserts at *HIS4*, showing digests used to detect recombination intermediates and products. One parent (P1) contains *ARG4* sequences with a VDE-recognition site (*arg4-VRS*), flanked by an nourseothricin-resistance module [*natMX*, (*Goldstein and McCusker, 1999*)] and the *Kluyveromyces lactis TRP1* gene [*KlTRP1*, (*Stark and Milner, 1989*)]; the other parent (P2) contains *ARG4* sequences with a mutant, uncuttable *VRS* site [*arg4-VRS103*, (*Nogami et al., 2002*) flanked by *URA3* and pBR322 sequences. Digestion with *Hin*dIII (H) and VDE (V) allows detection of crossovers (CO1 and CO2) and noncrossovers (NCO); digestion with *Hin*dIII alone allows detection of crossovers and DSBs. P2, CO1 and CO2 fragments are drawn only once, as they are the same size in *Hin*dIII digests as in *Hin*dIII + VDE digests. Right—representative Southern blots. *Hin*dIII-alone digests are probed with a fragment (probe 2) that hybridizes to the insert loci and to the native *ARG4* locus on chromosome VIII; this latter signal serves as a loading control (LC). Times after induction of meiosis that each sample was taken are indicated below each lane. (B) map of VDE-reporter inserts at *URA3* and representative Southern blots; details as in (A). Strain, insert and probe details are given in Materials and methods and *Supplementary file 1*.

The following figure supplement is available for figure 1:

*Figure 1 continued on next page*

*Figure 1 continued*

**Figure supplement 1.** Spo11-initiated events at the two insert loci.

cleaved by 7 hr after initiation of sporulation (*Figure 2A*), consistent with previous reports that VDE cuts very effectively (*Johnson et al., 2007*; *Neale et al., 2002*; *Terentyev et al., 2010*). Thus, in most cells, both sister chromatids are cut by VDE (*Gimble and Thorner, 1992*; *Neale et al., 2002*). In contrast, Spo11-DSBs infrequently occur at the same place on both sister chromatids (*Zhang et al., 2011*). While the consequences of this difference remain to be determined, we note that inserts at both *HIS4* and *URA3* are cleaved by VDE with equal frequency (*Figure 2A*). Thus, any effects due simultaneous sister chromatid-cutting should be equal at the two loci.

DSBs appeared and disappeared with similar timing at the two loci (*Figure 2B*), with measures of insert recovery (*Figure 2—figure supplement 1A*) and levels of interhomolog recombinants relative to cumulative VDE-DSB levels (*Figure 2—figure supplement 1B*) indicating that ~70% of VDE DSBs are repaired by interhomolog recombination. The remaining *VRS*-containing inserts appear to be lost, consistent with high levels of VDE activity preventing recovery of inter-sister recombinants. Thus, the two VDE recombination reporter inserts undergo comparably high levels of meiotic recombination initiation, regardless of the local intrinsic level of Spo11-initiated recombination.

When VDE-DSBs are repaired by interhomolog recombination, *VRS* sequences are converted to *VRS103*, and become resistant to digestion by VDE. We therefore used *Hin*dIII/VDE double digest to score recombinants that are resistant to VDE cleavage (*Figure 1*). Comparing the levels of such recombinants in VDE-expressing and *vde∆* strains indicates that Spo11-initiated events comprise only a few percent of the recombinants scored in VDE-expressing strains (*Figure 2C*, *Figure 1—figure supplement 1*, data not shown). VDE-initiated recombinants formed at high frequencies at both *HIS4* and *URA3*, and NCOs exceeded COs by approximately twofold at *HIS4* and threefold at *URA3* (*Figure 2C*). These values are within the range observed in genetic studies of Spo11-induced gene conversion in budding yeast (*Fogel et al., 1979*), but differ from the average of near-parity between NCOs and COs observed in molecular assays (*Lao et al., 2013*; *Martini et al., 2006*). This is

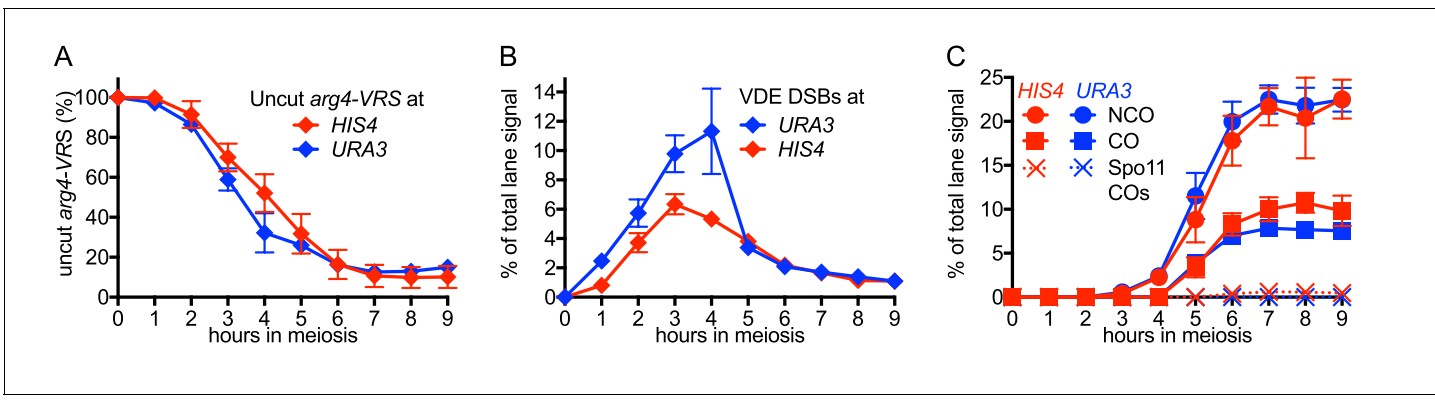

**Figure 2.** VDE-initiated recombination occurs at similar levels at the two insert loci. (A) Cumulative DSB levels are similar at the two insert loci. The fraction of uncut VRS-containing chromosomes (Parent 1) was determined by subtracting the amount of the NCO band in *Hin*dIII + VDE digests from the amount of the Parent 1 + NCO band in *Hin*dIII digests. (B) Non-cumulative VDE-DSB frequencies, measured as fraction of total lane signal, excluding loading controls, in *Hin*dIII digests. (C) Crossover (average of CO1 and CO2) and noncrossover frequencies, measured in *Hin*dIII-VDE digests. Solid lines—recombinants from cells expressing VDE; dashed lines—Spo11-initiated crossovers from *vde*⁻ strains, measured in *Hin*dIII-VDE digests and thus corresponding to VDE-resistant products (see also *Figure 1—figure supplement 1C*). Values are the average of two independent experiments; error bars represent range. Representative Southern blots are shown in *Figure 1* and *Figure 1—figure supplement 1C*.

The following figure supplement is available for figure 2:

**Figure supplement 1.** 70–80% of VDE-DSBs are repaired.

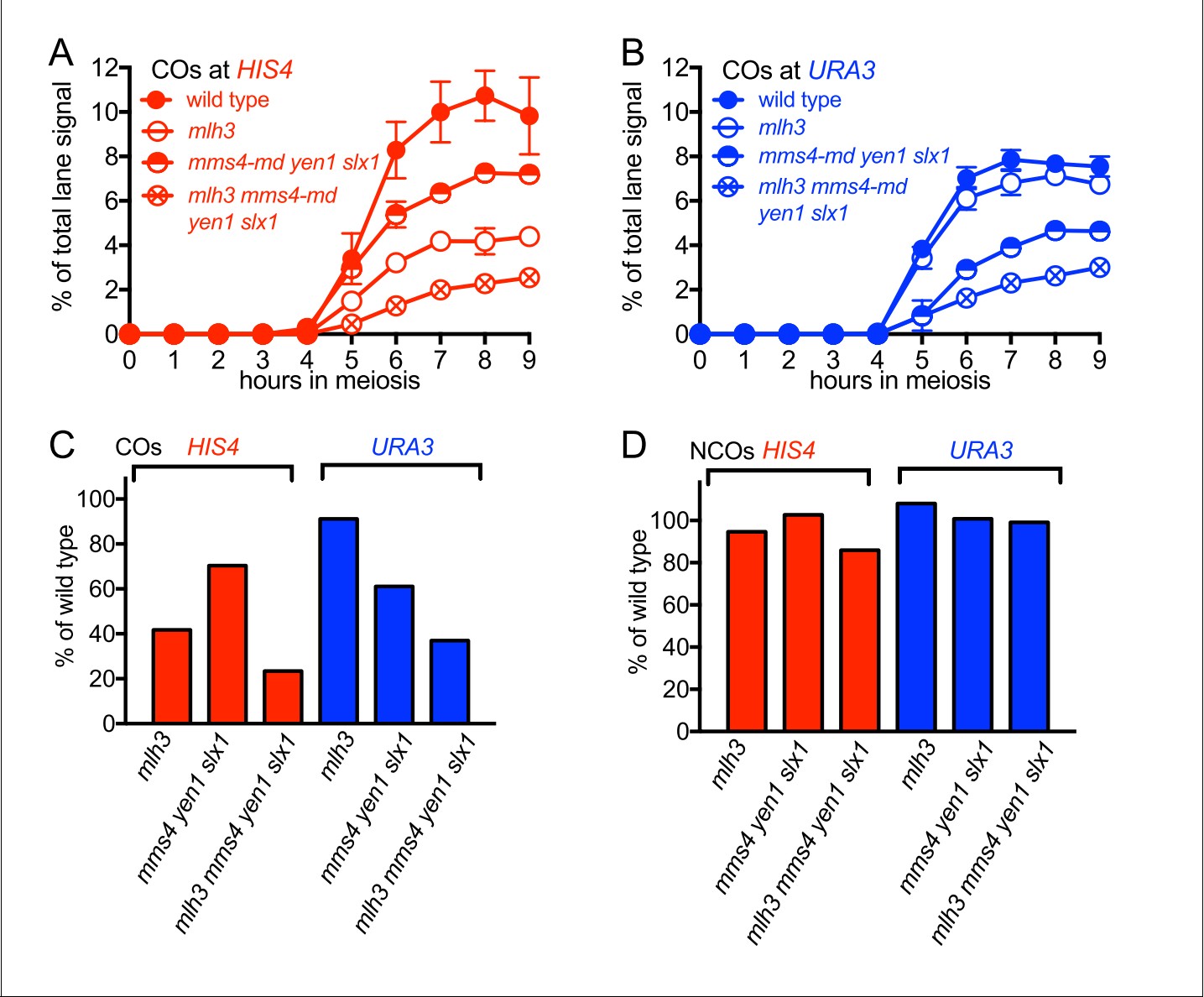

**Figure 3.** Different resolvase-dependence of crossover formation at the two insert loci. (A) Crossover frequencies (average of CO1 and CO2) measured as in *Figure 2C* from *HIS4* insert-containing mutants lacking MutLγ (*mlh3*), structure-selective nucleases (*mms4-md yen1 slx1*) or both resolvase activities (*mlh3 mms4-md yen1 slx1*). (B) Crossover frequencies in *URA3* insert-containing strains, measured as in panel A. Values are the average of two independent experiments; error bars represent range. (C) Final crossover levels (average of 8 and 9 hr values for two independent experiments), expressed as percent of wild type. Note that, in *mlh3* mutants, crossovers in *HIS4* inserts are reduced by nearly 60%, while crossovers in *URA3* inserts are reduced by less than 10%. (D) Final noncrossover levels, calculated as in C, expressed as percent of wild type. Representative Southern blots are in *Figure 3—figure supplement 2*.

The following figure supplements are available for figure 3:

**Figure supplement 1.** VDE-DSB and NCO frequencies in resolvase mutants.

**Figure supplement 2.** Southern blots of *Hin*dIII and *Hin*dIII-VDE digests of DNA from *HIS4* insert-containing strains (top) and from *URA3* insert-contaning strains (bottom).

consistent with earlier findings, that cutting both sister chromatids at a DSB site is associated with a reduced proportion of COs among repair products (*Malkova et al., 2000*).

## MutLγ makes different contributions to VDE-initiated CO formation at the two insert loci

While VDE-initiated recombination occurred at similar levels in inserts located at *HIS4* and at *URA3*, we observed a marked difference between the two loci, in terms of the resolvase-dependence of CO formation (*Figure 3*). At the *HIS4* locus, COs were reduced in *mlh3Δ* mutants, which lack MutLγ, by ~60% relative to wild type. In *mms4-md yen1Δ slx1Δ* mutants, which lack the three structure selective nucleases active during both meiosis and the mitotic cell cycle (SSNs, triple mutants hereafter called *ssn* mutants), COs were reduced by ~30%, and by ~75% in *mlh3 ssn* mutants. Thus, like Spo11-initiated COs, VDE-initiated COs in inserts at *HIS4* are primarily MutLγ-dependent, and less dependent on SSNs. In contrast, COs in inserts located at *URA3* were reduced by only ~ 10% in *mlh3*, by ~40% in *ssn* mutants, and by ~60% in *mlh3 ssn* mutants, so that the final level of residual COs was the same as at *HIS4*. Thus, SSNs make a substantially greater contribution to VDE-initiated CO formation at *URA3* than does MutLγ, and MutLγ's contribution becomes substantial only in the absence of SSNs.

At both insert loci, *ssn* and *mlh3 ssn* mutants accumulated DNA species with reduced electrophoretic mobility (*Figure 3—figure supplement 2*). These slower-migrating species contain branched DNA molecules, as would be expected for unresolved joint molecules (D. M., unpublished observations). Steady state VDE-DSB and final NCO levels were similar in all strains (*Figure 3D*, *Figure 3—figure supplement 1*), indicating that resolvases do not act during the initial steps of DSB repair, and consistent with most meiotic NCOs forming by mechanisms that do not involve Holliday junction resolution (*Allers and Lichten, 2001a*; *De Muyt et al., 2012*; *Sourirajan and Lichten, 2008*; *Zakharyevich et al., 2012*).

## Altered Hop1 occupancy in *pch2* mutants is associated with altered MutLγ– dependence of VDE-initiated COs

The marked MutLγ-dependence and –independence of VDE-initiated COs in inserts at *HIS4* and at *URA3*, respectively, are paralleled by the levels of occupancy of the meiotic axis proteins Hop1 and Red1 (*Panizza et al., 2011*; *Figure 4A*, *Figure 4—figure supplement 1A*). To test further the suggestion that differential Hop1 occupancy is correlated with differences in CO formation at these loci, we examined the resolvase-dependence of VDE-initiated COs in *pch2Δ* mutants. Pch2 is a conserved AAA+ ATPase that maintains the nonuniform pattern of Hop1 occupancy along meiotic chromosomes (*Börner et al., 2008*; *Joshi et al., 2009*). The different Hop1 occupancies seen in wild type were preserved early in meiosis in *pch2Δ* mutants (*Figure 4A*, *Figure 4—figure supplement 1A*), consistent with previous findings that, in *pch2* cells, Spo11-DSB patterns are not altered in most regions of the genome (*Vader et al., 2011*). By contrast, at later times (4–5 hr after initiation of meiosis), *pch2Δ* mutants displayed reduced Hop1 occupancy at *HIS4*, more closely approaching the lower occupancy levels seen throughout meiosis at *URA3* (*Figure 4A*; *Figure 4—figure supplement 1A*).

The altered Hop1 occupancy in *pch2Δ* was accompanied by altered resolvase contributions to VDE-initiated COs (*Figure 4B,C,D*). MutLγ contributions decreased at *HIS4* and increased at *URA3*, and the majority of COs were MutLγ-independent at both insert loci. In contrast, SSN contributions increased slightly at *HIS4*, and remained unchanged at *URA3*. Thus, in *pch2Δ* mutants, the similarity of Hop1 occupancy at later times in meiosis is paralleled by a shift towards more similar contributions of MutLγ to VDE-initiated COs at *HIS4* and *URA3*. Finally, VDE-induced DSB dynamics and NCO levels were similar in *PCH2* and *pch2Δ* strains, except that NCO levels at both loci were reduced in *pch2Δ mms4-md yen1Δ slx1Δ*, suggesting a greater role for SSNs in NCO formation in the absence of Pch2 (*Figure 4—figure supplement 1B,C*).

## Spo11-DSBs promote VDE-initiated, MutLγ-dependent COs

All of the experiments reported above used cells with wild-type levels of Spo11-DSBs. While VDE-DSBs form at similar levels and timing in *SPO11* and *spo11* mutant cells (*Johnson et al., 2007*; *Neale et al., 2002*; *Terentyev et al., 2010*), features of VDE-DSB repair, including the extent of end

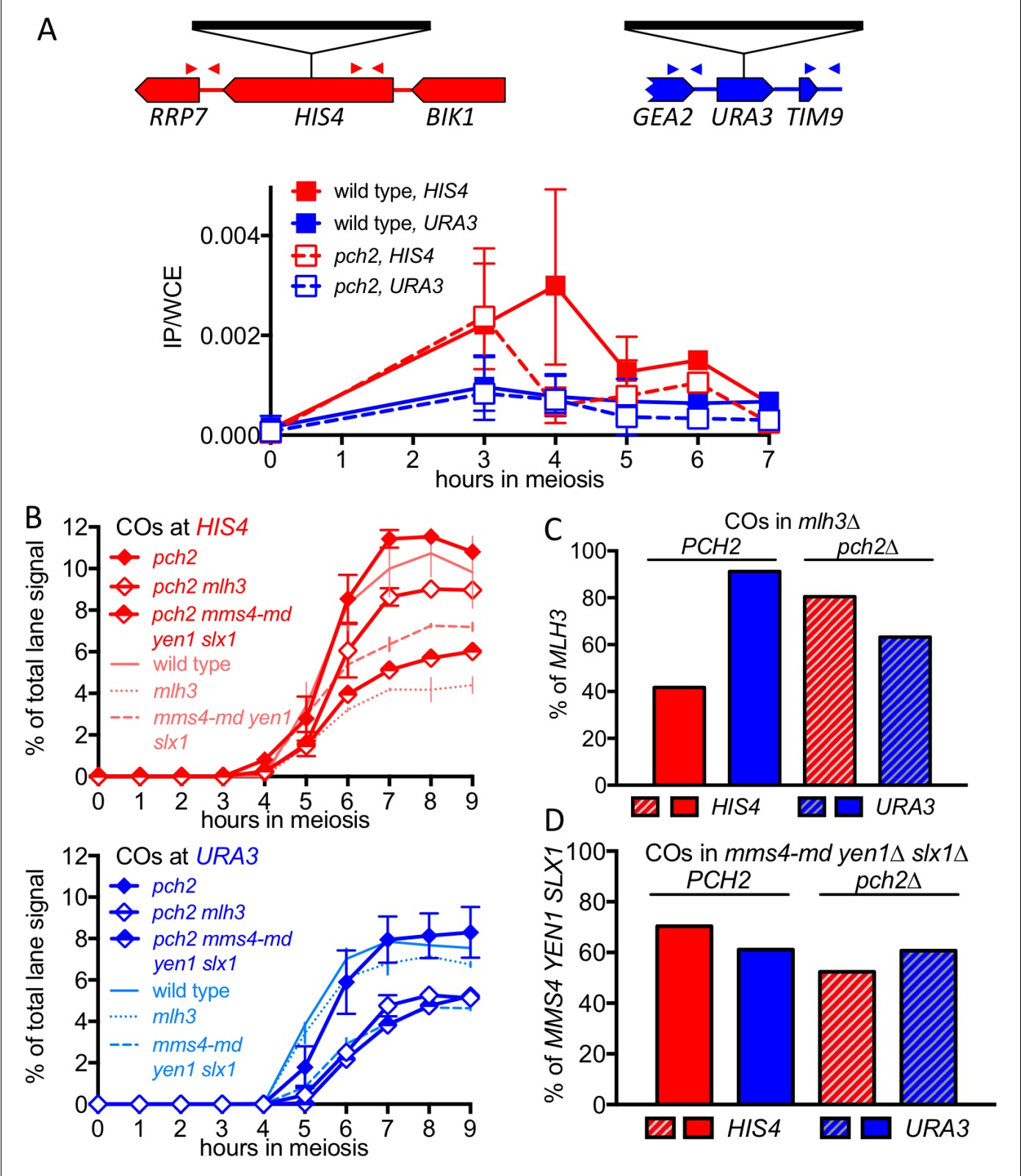

**Figure 4.** *pch2Δ* mutants display altered Hop1 occupancy and crossover MutLγ-dependence. (**A**) Hop1 occupancy at insert loci, determined by chromatin immunoprecipitation and quantitative PCR. Top—cartoon of insert loci, showing the location of primer pairs used. Bottom—relative Hop1 occupancy, expressed as the average ratio of immunoprecipitate/input extract for both primer pairs (see Materials and methods for details). Values are the average of two independent experiments; error bars represent range. (**B**) VDE-initiated CO frequencies measured as in **Figure 2C** at *HIS4* (top) and

*Figure 4 continued on next page*

*Figure 4 continued*

URA3 (bottom) in *pch2Δ* (solid diamonds), *pch2Δ mlh3Δ* (open diamonds), and *pch2Δ mms4-md yen1 slx1* (half-filled diamonds) mutants. Crossovers from wild type (solid line), *mlh3Δ* (dotted line) and *mms4-md yen1 slx1* mutants (dashed line) from **Figure 3** are shown for comparison. Values are from two independent experiments; error bars represent range. Representative Southern blots are in **Figure 4—figure supplement 2**. (C) Extent of CO reduction in *mlh3Δ* mutants, relative to corresponding *MLH3* strains. (D) Extent of CO reduction in *mms4-md yen1 slx1* (*ssn*) mutants, relative to corresponding *MMS4 YEN1 SLX1* strains. For both (C) and (D), *PCH2* genotype is indicated at the top; values are calculated as in **Figure 3C**.

The following figure supplements are available for figure 4:

**Figure supplement 1.** Hop1 occupancy at non-insert loci, DSBs and NCOs in *pch2Δ* mutants.

**Figure supplement 2.** Southern blots of *Hind*III and *Hind*III-VDE digests of DNA from *HIS4* insert-containing strains (top) and from *URA3* insert-contaning strains (bottom).

resection, are strongly influenced by the presence or absence of Spo11-DSBs (*Neale et al., 2002*). To determine if other aspects of VDE-initiated recombination are also affected, we examined VDE-initiated recombination in a catalysis-null *spo11-Y135F* mutant, hereafter called *spo11*. In *spo11* mutants, VDE-DSB dynamics and NCO formation were similar in inserts at *HIS4* and *URA3*, were comparable to those seen in wild type (*Figure 5—figure supplement 1*), and were independent of HJ resolvase activities (*Figure 5—figure supplement 1*). In contrast, the absence of Spo11-DSBs substantially reduced VDE-induced COs, resulting in virtually identical CO timing and levels at the two loci (*Figure 5A*). Unlike the ~60% reduction in COs seen at *HIS4* in *SPO11 mlh3Δ* (*Figure 3C*), final CO levels were similar in *spo11 mlh3Δ* and *spo11 MLH3* strains, at both *HIS4* and *URA3*, and similar CO reductions were observed at both loci in *spo11 ssn* mutants (*Figure 5B,C*). Thus, processes that depend on Spo11-DSBs elsewhere in the genome are important to promote VDE-initiated COs, and appear to be essential for MutLγ-dependent CO formation.

## Discussion

### Local chromosome context influences meiotic CO formation

We examined the contribution of different Holliday junction resolvases to VDE-initiated CO-formation in recombination reporter inserts at two loci, *HIS4* and *URA3*, which are 'hot' and 'cold', respectively, for Spo11-inititiated recombination and for occupancy by the meiotic chromosome axis proteins, Hop1 and Red1. VDE-initiated COs at *HIS4* are similar to those initiated by Spo11, in that most depend on MutLγ. In contrast, VDE-initiated COs at the 'cold' locus, *URA3*, more closely resemble mitotic COs, which are independent of MutLγ, but are substantially dependent on SSNs (*Ho et al., 2010*; *Ira et al., 2003*; *Muñoz-Galván et al., 2012*). Locus-dependent differences in MutLγ-dependence are reduced in *pch2Δ* mutants, as are differences in Hop1 occupancy at later times in meiosis I prophase. Based on these findings, we suggest that local chromosome context exerts an important influence on the biochemistry of CO formation during meiosis, and that factors responsible for creating DSB-hot and -cold domains also create corresponding domains where different DSB repair pathways are dominant. An attractive hypothesis (*Figure 6*) is that regions enriched for meiosis-specific axial element proteins create a chromosomal environment that promotes meiotic DSB formation, limits inter-sister recombination, preferentially loads ZMM proteins (*Joshi et al., 2009*; *Serrentino et al., 2013*), and is required for recruitment of MutLγ. In such regions, where most Spo11-dependent events occur, recombination intermediates will have a greater likelihood of being captured by axis-associated ZMM proteins, and consequently being resolved as COs by MutLγ. Regions with lower axial element protein enrichment are less likely to recruit ZMM proteins and MutLγ; DSB repair and CO formation in these regions are more likely to involve non-meiotic mechanisms. In short, the meiotic genome can be thought of as containing two types of environment: meiotic axis protein-enriched regions, where 'meiotic' recombination pathways predominate; and meiotic axis protein-depleted regions, in which recombination events more closely resemble those seen in mitotic cells.

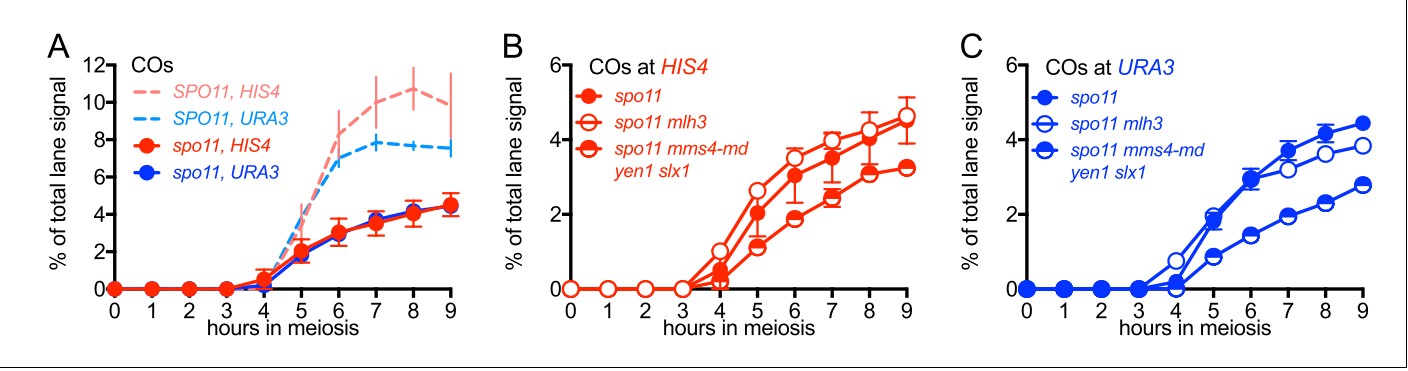

**Figure 5.** VDE-initiated COs are reduced and are MutLγ-independent in the absence of Spo11 activity. (**A**) VDE-initiated crossover frequencies, measured as in *Figure 2C* in *spo11-Y135F* strains (dark solid lines) in inserts at *HIS4* (red) and at *URA3* (blue). Data from the corresponding *SPO11* strains (dotted lines, from *Figure 2C*) are presented for comparison. (**B**) COs in *HIS4* inserts in *spo11* strains that are otherwise wild-type (*spo11*) or lack either Mutlγ or structure-selective nucleases. (**C**) As in **B**, but with inserts at *URA3*. Values are from two independent experiments; error bars represent range. Representative Southern blots are in *Figure 5—figure supplement 2*.

The following figure supplements are available for figure 5:

**Figure supplement 1.** DSBs and recombinant products in *spo11* strains.
**Figure supplement 2.** Southern blots of *Hin*dIII and *Hin*dIII-VDE digests of DNA from *spo11* strains with inserts at *HIS4* (top) and at *URA3* (bottom).

The observation that some COs at *HIS4* are SSN-dependent, even though most are MutLγ-dependent (*Figure 3*), indicates that this division is not absolute. In addition, it is important to keep in mind that ChIP-based values for meiotic axis protein-enrichment and molecular measures of CO resolvase-dependence are both population-based averages, and do not detect cell-to-cell heterogeneity. It is possible that meiotic axis protein enrichment at *HIS4* varies across a population, and most SSN-dependent COs form in cells where *HIS4* is not meiotic axis protein-enriched. Alternatively, it is possible that meiotic axis protein enrichment at *HIS4* is uniform across a population, but that MutLγ is recruited to JMs with less than unit efficiency, and that when MutLγ is not recruited, SSNs resolve JMs. Finally, it is important to recognize that, while meiotic axis protein occupancy is an attractive candidate as a determinant of resolvase contributions to VDE-induced CO formation, other explanations are possible. It is possible that the associations seen at *HIS4* and *URA3*, rather than being directly causative, reflect another underlying aspect of meiotic chromosome structure or function, and that other differences between these two loci cause the observed differences in resolvase usage.

While the current study is the first to directly query the effect of chromosome context on JM resolution, others have obtained results that are consistent with an effect of local chromosome context on meiotic DSB repair. Malkova and coworkers used the HO endonuclease to initiate recombination in meiotic cells at *LEU2*, also a 'hot' locus (*Panizza et al., 2011*; *Wu and Lichten, 1995*). The resulting COs were dependent on Msh4, a ZMM protein, to the same degree as are Spo11-induced COs, suggesting that these nuclease-induced COs at the axis enriched *LEU2* locus were the products of ZMM/MutLγ-dependent JM resolution (*Malkova et al., 2000*). *Serrentino et al. (2013)* showed that enrichment for the budding yeast ZMM protein, Zip3, at DSB sites is correlated with interhomolog CO levels. Specialized chromosome elements also impact meiotic recombination in budding yeast: COs are differentially reduced relative to NCOs near telomeres (*Chen et al., 2008*); and interhomolog recombination is inhibited near centromeres (*Chen et al., 2008*; *Lambie and Roeder, 1988*, *1986*; *Vincenten et al., 2015*). Locus-specific differences in CO/NCO ratios also have been observed in mouse meiosis (*de Boer et al., 2015*), locus-specific differences in partner choice have been reported in *S. pombe* (*Hyppa and Smith, 2010*), and crossover suppression by centromeres is observed in many species (*Talbert and Henikoff, 2010*).

Consistent with the suggestion that different meiotic recombination uses different mechanisms in different regions, the meiotic genome also appears to contain regions that differ in terms of the

response to DNA damage. Treatment of meiotic yeast cells with phleomycin, a DSB-forming agent, triggers Rad53 phosphorylation, as it does in mitotic cells, while Spo11-DSBs do not (*Cartagena-Lirola et al., 2008*). This suggests that Spo11-DSBs form in an environment that is refractory to Rad53 recruitment and modification, but that there also are environments where exogenously-induced damage can trigger the mitotic DNA damage response. In light of this suggestion, it is interesting that the meiotic defects of *spo11* mutants in a variety of organisms are often only partially rescued by DSBs caused by exogenous agents (*Bowring et al., 2006*; *Celerin et al., 2000*; *Dernburg et al., 1998*; *Loidl and Mochizuki, 2009*; *Pauklin et al., 2009*; *Storlazzi et al., 2003*; *Thorne and Byers, 1993*). While other factors may be responsible for the limited rescue observed, we suggest that it reflects the random location of exogenously-induced DSBs, with only a subset

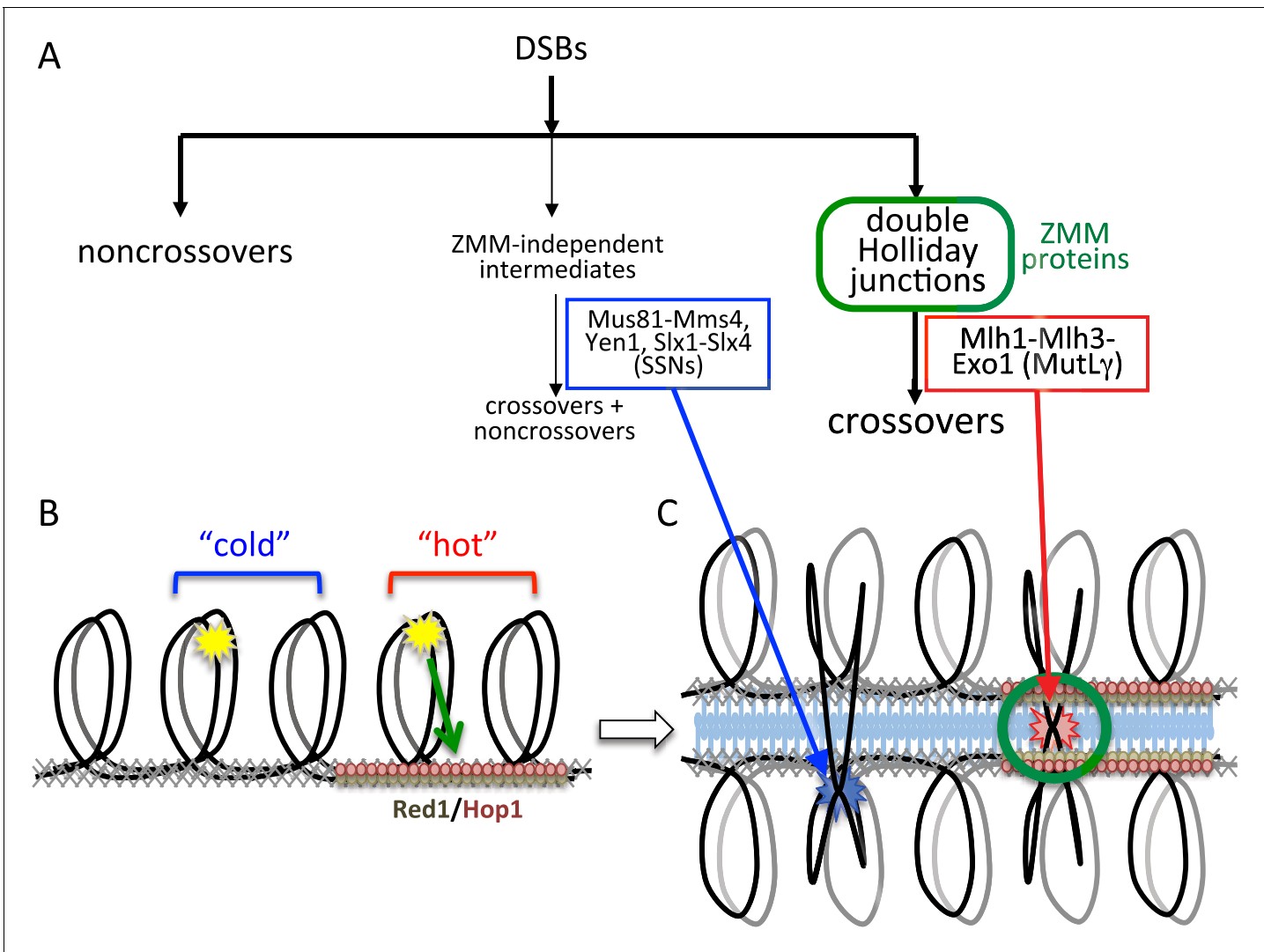

**Figure 6.** Different resolvase functions in different genome domains. (**A**) Early crossover decision model for meiotic recombination (*Bishop and Zickler, 2004*; *Hollingsworth and Brill, 2004*) illustrating early noncrossover formation, a major pathway where recombination intermediates form in the context of ZMM proteins and are resolved by MutLγ to form crossovers, and a minor pathway where ZMM-independent intermediates are resolved by SSNs as both crossovers and noncrossovers. (**B**) Division of the meiotic genome into meiotic axis-protein-enriched 'hot' domains (red) that are enriched for Red1 and Hop1, and 'cold' domains where Red1 and Hop1 are depleted. VDE DSBs (yellow stars) can be directed to form efficiently in either domain, but only VDE DSBs that form in 'hot' domains can be recruited to the meiotic axis. (**C**) DSBs in 'hot' domains can form joint molecules (red star) in the context of ZMM proteins and the synaptonemal complex, and thus can be resolved by MutLγ-dependent activities. DSBs in 'cold' domains form joint molecules (blue star) outside of this structural context, and are resolved by MutLγ-independent activities.

forming in regions where repair is likely to form interhomolog COs that promote proper homolog segregation.

## The interplay of resolvase activities is chromosome context-dependent

Although we observe marked differences in the contributions of different resolvases to VDE-induced CO formation at *HIS4* and at *URA3*, there is no absolute demarcation between MutLγ and SSN activities at the two loci. At *HIS4*, where MutLγ predominates, *ssn* mutants still display a modest reduction in VDE-initiated COs when MutLγ is active, but an even greater relative reduction in the absence of MutLγ. These findings are consistent with previous studies suggesting that, in the absence of MutLγ, SSNs serve as a back-up that resolves JMs to produce both COs and NCOs (*Argueso et al., 2004*; *De Muyt et al., 2012*; *Zakharyevich et al., 2012*). Our current data indicate that the converse may also be true, since at *URA3*, MutLγ appears to make a greater contribution to CO formation in the absence of SSNs than in their presence. However, in our studies, JMs are more efficiently resolved in *mlh3Δ* mutants than in *ssn* mutants, which display persistent unresolved JMs. Therefore, if MutLγ acts as a back-up resolvase, it can do so in only a limited capacity, possibly reflecting a need for a specific chromosome structural context in which MutLγ can be efficiently loaded and activated. The absence of such a meiosis-specific chromosome context may explain why MutLγ does not appear to contribute to CO formation during the mitotic cell cycle (*Ira et al., 2003*), although lower *MLH3* expression in mitotic cells (*Primig et al., 2000*) may also reduce its contribution.

Both VDE-induced and Spo11-induced COs form at significant frequencies in *mlh3Δ ssn* mutants, which lack all four of the HJ resolvase activities thought to function during meiosis (*Figure 3*; *Argueso et al., 2004*; *Zakharyevich et al., 2012*). These residual crossovers may reflect the activity of a yet-unidentified JM resolvase; they may also reflect the production of half-crossovers by break-induced replication (*Ho et al., 2010*; *Kogoma, 1996*; *Llorente et al., 2008*) or by other mechanisms that do not involve dHJ-JM formation and resolution (*Ivanov and Haber, 1995*; *Mazón et al., 2012*; *Muñoz-Galván et al., 2012*). Alternatively, long-tract NCO gene conversion events that include flanking heterologous sequences might be responsible for the products, scored in our molecular assays as COs, that are independent of both MutLγ and SSNs.

## Genome-wide Spo11-DSBs promote VDE-initiated COs and are required for chromosome context-dependent differentiation of VDE DSB repair

In catalysis-null *spo11-Y135F* mutants, most VDE-DSBs are repaired by interhomolog recombination (*Figure 5*, *Figure 5—figure supplement 2*), indicating that a single DSB can efficiently search the meiotic nucleus for homology. However, VDE-promoted COs are substantially reduced in *spo11* mutants (*Figure 5*), as has been observed with HO endonuclease-induced meiotic recombination (*Malkova et al., 2000*). Moreover, in *spo11* mutants, virtually all VDE-initiated COs are MutLγ-independent (*Figure 5*, *Figure 5—figure supplement 2*), and thus more closely resemble COs that form in mitotic cells. Because patterns of Hop1 occupancy are not markedly altered in *spo11* mutants (Franz Klein, personal communication), these findings indicate that, in addition to the local effects of meiotic chromosome structure suggested above, meiotic CO formation is affected by processes that require Spo11-DSBs elsewhere in the genome.

Meiotic DSB repair occurs concurrently with homolog pairing and synapsis (*Börner et al., 2004*; *Padmore et al., 1991*), and efficient homolog synapsis requires wild-type DSB levels (*Henderson and Keeney, 2004*), indicating that multiple interhomolog interactions along a chromosome are needed for stable homolog pairing. To account for the reduced levels and MutLγ-independence of VDE-initiated COs in *spo11* mutants, we suggest that a single VDE-DSB is not sufficient to promote stable homolog pairing, and that additional DSBs along a chromosome are needed to promote stable homolog pairing, which in turn is needed to form ZMM protein-containing structures that stabilize JMs and recruit MutLγ. However, the 140–190 Spo11-DSBs that form in each meiotic cell (*Buhler et al., 2007*) are also expected to induce a nucleus-wide DNA damage-response, and to compete with other DSBs for repair activities whose availability is limited, and both have the potential to alter recombination biochemistry at VDE-DSBs (*Johnson et al., 2007*; *Neale et al., 2002*). Thus, while we believe it likely that defects in homolog pairing and synapsis are responsible for the

observed impact of *spo11* mutation on VDE-initiated CO formation, it remains possible that it is due to changes in DNA damage signaling, repair protein availability, or in other processes that are affected by global Spo11-DSB levels.

## Concluding remarks

We have provided evidence that structural features of the chromosome axis, in particular the enrichment for meiosis-specific axis proteins, create a local environment that directs recombination to 'meiotic' biochemical pathways. In the remainder of the genome, biochemical processes more typical of mitotic recombination function. In other words, the transition to meiosis from the mitotic cell cycle does not involve a global inhibition of 'mitotic' recombination pathways. These 'mitotic' mechanisms remain active in the meiotic nucleus, and can act both in recombination events that occur outside of the local 'meiotic' structural context, and in recombination in *spo11* mutants. It is well established that local chromosome context influences the first step in meiotic recombination, Spo11-catalyzed DSB formation (*Panizza et al., 2011*; *Prieler et al., 2005*). Our work shows that it also influences the last, namely the resolution of recombination intermediates to form COs. It will be of considerable interest to determine if other critical steps in meiotic recombination, such as choice between sister chromatid and homolog as a DSB repair partner, or the choice between NCO and CO outcomes, are also influenced by local aspects of interstitial chromosome structure.

In the current work, we focused on correlations between local enrichment for the meiosis-specific axis protein Hop1 and Holliday junction resolution activity during CO formation. Other HORMA domain proteins, including HIM-3 and HTP-1/2/3 in *C. elegans*, ASY3 in *A. thaliana* and HORMAD1/2 in *M. musculus*, also have been reported to regulate recombination and homolog pairing (*Ferdous et al., 2012*; *Fukuda et al., 2010*; *Kim et al., 2014*; *Wojtasz et al., 2009*), suggesting that HORMA domain proteins may provide a common basis for the chromosome context-dependent regulation of meiotic recombination pathways in eukaryotes.

# Materials and methods

## Yeast strains

All yeast strains are of SK1 background (*Kane and Roth, 1974*), and were constructed by standard genetic crosses or by direct transformation. Genotypes and allele details are given in *Supplementary file 1*. Recombination reporter inserts with *arg4-VRS103* contain a 73nt *VRS103* oligonucleotide containing the mutant VDE recognition sequence from the *VMA1-103* allele (*Fukuda et al., 2007*; *Nogami et al., 2002*) inserted at the *Eco*RV site in *ARG4* coding sequences within a pBR322-based plasmid with *URA* and *ARG4* sequences, inserted at the *URA3* and *HIS4* loci, as described (*Wu and Lichten, 1995*). Recombination reporter inserts with the cleavable *arg4-VRS* (*Neale et al., 2002*) were derived from similar inserts but with flanking repeat sequences removed, to prevent repair by single strand annealing (*Pâques and Haber, 1999*). This was done by replacing sequences upstream and downstream of *ARG4* with *natMX* (*Goldstein and McCusker, 1999*) and *K. lactis TRP1* sequences (*Stark and Milner, 1989*) respectively (see *Supplementary file 1* legend for details). The resulting *arg4-VRS* and *arg4-VRS103* inserts share 3.077 kb of homology.

VDE normally exists as an intein in the constitutively-expressed *VMA1* gene (*Gimble and Thorner, 1993*), resulting in low levels of DSB formation in presporulation cultures (data not shown), probably due to small amounts VDE incidentally imported to the nucleus during mitotic growth (*Nagai et al., 2003*). To further restrict VDE DSB formation, strains were constructed in which *VDE* expression was copper-inducible. These strains contain the *VMA1-103* allele (*Nogami et al., 2002*), which provides wild type *VMA1* function, but lacks the VDE intein and is resistant to cleavage by VDE. To make strains in which *VDE* expression was copper-inducible, *VDE* coding sequences on an *Eco*RI fragment from pY2181 (*Nogami et al., 2002*; a generous gift from Dr. Satoru Nogami and Dr. Yoshikazu Ohya) were inserted downstream of the CUP1 promoter in plasmid pHG40, which contains the *kanMX* selectable marker and a ~1 kb *CUP1* promoter fragment (*Jin et al., 2009*), to make pMJ920, which was then integrated at the *CUP1* locus.

## Sporulation

Yeast strains were grown in buffered liquid presporulation medium and shifted to sporulation medium as described (**Goyon and Lichten, 1993**), except that sporulation medium contained 10 uM CuSO₄ to induce *VDE* expression. All experiments were performed at 30°C.

## DNA extraction and analysis

Genomic DNA was prepared as described (**Allers and Lichten, 2000**). Recombination products were detected on Southern blots containing genomic DNA digested with *Hind*III and *VDE* (*PI-Sce*I, New England Biolabs), using specific buffer for *PI-Sce*I. Samples were heated to 65°C for 15 min to disrupt VDE-DNA complexes before loading; gels contained 0.5% agarose in 45 mM Tris Borate + 1 mM EDTA (1X TBE) and were run at 2 V/cm for 24–25 hr. DSBs were similarly detected on Southern blots, but were digested with *Hind*III alone as previously described (**Goldfarb and Lichten, 2010**), and electrophoresis buffer was supplemented with 4 mM MgCl₂. Gels were transferred to membranes and hybridized with radioactive probe as described (**Allers and Lichten, 2001a**, **2001b**), and were imaged and quantified using a Fuji FLA-5100 phosphorimager and ImageGauge 4.22 software. *Hind*III-VDE gel blots were probed with *ARG4* sequences from −430 to +63 nt relative to *ARG4* coding sequences (Probe 1, **Figure 1**). To correct for the low level of uncut VDE sites present in all VDE digests (see **Figure 1**), NCO frequencies measured from these digests were adjusted by subtracting the frequency of apparent NCOs in 0 hr samples. *Hind*III gel blots were probed with sequences from the *DED81* gene (+978 to +1650 nt relative to *DED81* coding sequence), which is immediately upstream of *ARG4* (Probe 2, **Figure 1**). Digests of *sae2Δ* strains (**Figure 1—figure supplement 1**) were probed with nt 3149–4351 of pBR322.

## Chromatin immunoprecipitation and quantitative PCR

Cells were formaldehyde-fixed by adding 840 μl of a 36.5–38% formaldehyde solution (Sigma) to 30 ml of meiotic cultures, incubating for 15 min at room temperature, and quenched by the addition of glycine to 125 mM. Cells were harvested by centrifugation, resuspended in 500 μl lysis buffer (**Strahl-Bolsinger et al., 1997**) except with 1 mg/ml Bacitracin and complete protease inhibitor cocktail (one tablet/10 ml, Roche 04693116001) as protease inhibitors, and cells were lysed at 4°C via 10 cycles of vortexing on a FastPrep24 (MP Medical) at 4 M/sec for 40 s, with 5 min pauses between runs. Lysates were then sonicated to yield an average DNA size of 300 bp and clarified by centrifugation at 21,130 RCF for 20 min. 1/50th of the sample (10 μl) was removed as input, and 2 μl of anti-Hop1 (a generous gift from Nancy Hollingsworth) was added to the remainder (~490 μl) and incubated with gentle agitation overnight at 4°C. Antibody complexes were purified by addition of 20 μl of 50% slurry of Gammabind G Sepharose beads (GE Healthcare 17088501), with further incubation for 3 hr at 4°C, followed by pelleting at 845 RCF for 30 s. Beads were then processed for DNA extraction (**Blitzblau et al., 2012**; Viji Subramanian and Andreas Hochwagen, personal communication). Beads were washed with 1 ml lysis Buffer and once each with 1 ml high salt lysis buffer (same as lysis buffer except with 500 mM NaCl), 1 ml ChIP wash buffer (10 mM Tris, 0.25M LiCl, 0.5% NP-40, 0.5% sodium deoxycholate, 1 mM EDTA) and 1 mL 10 mM Tris, 1 mM EDTA; all washes were done for 5 min at room temperature. DNA was then eluted from beads by adding 100 ml 10 mM Tris, 1 mM EDTA, 1% SDS and incubating at 65°C for 15 min. Beads were then pelleted by a short spin at 16,363 RCF and the eluate transferred to a fresh tube. Beads were washed again in 150 ml 10 mM Tris, 1 mM EDTA, 0.67% SDS, mixed and pelleted again. Both eluates were pooled and crosslinks reversed for both immunoprecipitated (IP) and input samples by incubating overnight at 65°C. 250 ml 10 mM Tris 1 mM EDTA, 4 ml 5 mg/ml linear acrylamide (20 mg) and 5 ml 20 mg/ml Proteinase K (100 mg) was added, and samples were incubated at 37°C for 30 min for immunoprecipitates and 2 hr for input samples. 44 μl 5M LiCl was then added to immunoprecipitates, and DNA was precipitated by adding 1 ml ice cold ethanol, incubating at −20°C for 20 min, and centrifugation at 21,130 RCF for 20 min. For input samples, 44 ml 5M LiCl was added, followed by extraction with an equal volume of phenol:chloroform:isoamyl alcohol (25:24:1) and centrifugation at 16,363 RCF for 10 min. The aqueous layer was transferred to a fresh tube and DNA was precipitated from input samples as with immunoprecipitate samples.

qPCR analysis of purified DNA from input and immunoprecipitated samples used primer pairs that amplify two regions: chromosome III coordinates 65350–65547 and 68072–68271,

Saccharomyces Genome Database, flanking the *HIS4* gene, and chromosome V coordinates 115119–115317 and 117728–117922, flanking the *URA3* gene (see *Figure 1—figure supplement 1*). Chromosome coordinates are from the *Saccharomyce cerevisiae* reference genome (*Engel et al., 2014*). Primers and genomic DNA from input and immunoprecipitated samples were mixed with iQ SYBR green supermix (Biorad) and analyzed using a Biorad iCycler.

## Source data
Numerical values underlying all graphs are contained in *Supplementary file 2*.

## Acknowledgements
We thank Robert Shroff, Anuradha Sourirajan, Satoru Nogami, Yoshikazu Ohya, and Nancy Hollingsworth for generous gifts of strains and reagents, Viji Subramanian and Andreas Hochwagen for technical advice, and Dhruba Chattoraj, Julie Cooper, and Alex Kelly for comments on the manuscript. This work was supported by The University of Sheffield and the Intramural Research Program of the NIH through the Center for Cancer Research, National Cancer Institute.

## Additional information

### Funding

| Funder | Grant reference number | Author |
| --- | --- | --- |
| National Cancer Institute | Intramural Research Program of the NIH through the Center for Cancer Research at the National Cancer Institute | Michael Lichten |
| University of Sheffield | Graduate tuition grant | Darpan Medhi |

The funders had no role in study design, data collection and interpretation, or the decision to submit the work for publication.

### Author contributions
DM, Conception and design, Acquisition of data, Analysis and interpretation of data, Drafting or revising the article; ASHG, ML, Conception and design, Analysis and interpretation of data, Drafting or revising the article

### Author ORCIDs
Michael Lichten, http://orcid.org/0000-0001-9707-2956

## Additional files

### Supplementary files
• Supplementary file 1. Yeast strains.

• Supplementary file 2. Primary data for graphs in all figures.

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
