## [Decision Letter]

Thank you for submitting your article "Local chromosome context is a major determinant of crossover pathway biochemistry during budding yeast meiosis" for consideration by *eLife*. Your article has been favorably evaluated by Kevin Struhl (Senior Editor) and three reviewers, one of whom, Bernard de Massy (Reviewer #1), is a member of our Board of Reviewing Editors. The following individual involved in review of your submission has agreed to reveal their identity: Joao Matos (Reviewer #3).

The reviewers have discussed the reviews with one another and the Reviewing Editor has drafted this decision to help you prepare a revised submission.

Summary:

In this study Medhi et al. have analyzed the contribution of different Holliday junction resolvases to VDE-initiated CO-formation in recombination reporters inserted at HIS4 and URA3 loci. One known and documented difference between these two loci is the occupancy for the chromosome axis protein Hop1, high at HIS4 and low at URA3. The authors find that VDE-initiated COs at HIS4 are strongly dependent on MutLγ, similar to those initiated by Spo11. VDE-induced COs at URA3 are more dependent on SSNs. The authors also show that *pch2∆* mutants, which have smaller differences in Hop1 occupancy between HIS4 and URA3, display smaller locus-dependent differences in formation of COs by MutLγ. The authors conclude that local chromosome context is important for the biochemistry of CO formation during meiosis.

The data is of high quality and the striking differences between insertions at the HIS4 and URA3 loci are clearly documented. These results provide significant, interesting and novel insights into the regulation of meiotic recombination by properties of chromosome structure.

Essential revisions:

In order to fully validate the interpretations, some additional experiments are needed using the tools that the authors have used in the present experiments. In addition, the authors should not over interpret the data which is based on the comparison of two loci (HIS4 and URA3) with a correlation with HOP1 occupancy. There are a large number of possible differences between these loci aside from the ones the authors focus on, and it remains possible that the direction of the correlation seen with these two loci remains purely coincidental. Pch2 mutant fits the expectation but absence of Pch2 has likely several other consequences. The Abstract and conclusions should be modified accordingly. Use of SEM should me revised.

1) In order to evaluate the generality of their findings (see general comment above), the authors could compare published ChIP-seq data of Hop1 with available genome wide recombination maps from resolvase mutants.

2) One important experiment missing in this paper is to demonstrate the requirement for *ssn* in *pch2* mutant and thus to analyze intermediates, COs and NCOs in *pch2 mms4 yen1 slx1* mutant.

3) Since there are only two replicate datasets for several analyses, error bars should show range rather than SEM for the time-course plots. Bar graphs should be replaced with univariate scatter plots, but if the authors wish to retain the bar graphs, then error bars should show range, not SEM. This paper in PLoS Biology provides an excellent discussion of pitfalls for bar graphs and suggests other strategies for data display: http://journals.plos.org/plosbiology/article?id=10.1371/journal.pbio.1002128

4) The authors should explicitly indicate that in most meiosis VDE cuts both sister chromatids and that the consequences of this on pathway choices are unknown.

5) In the subsection “Local chromosome structure influences meiotic CO formation”, second paragraph: There is little or no mention of prior studies of context-dependence for crossover-noncrossover likelihood (Mancera et al. 2008; Serrentino et al. 2013; deBoer et al. 2015) or interhomolog vs. intersister partner choice (Hyppa & Smith 2010; Fowler et al. 2014). The Serrentino paper is mentioned briefly in passing in the preceding paragraph, but I doubt a reader would realize from this mention that paper had documented differences in crossover vs. noncrossover outcome between different loci. There are also studies documenting the different recombination behavior for DSBs within pericentromeric regions (Chen et al. 2008; Vincenten et al. 2015). These prior studies should be discussed in comparison to the context dependence documented here. (This is also relevant to the statement in the subsection “Concluding remarks”, at the end of the first paragraph.)

*Reviewer #1:*

Taking advantage of DSBs induced by the endonuclease VDE in *S. cerevisiae* together with genetic and molecular analysis of recombination intermediates during meiosis, Medhi et al. show distinct requirements for resolution of recombination intermediates in correlation with chromosomal context, i.e. the enrichment for the axis protein Hop1: in a Hop1-enriched region, MutLg (Mlh1, Mlh3, Exo1) is the main resolvase involved, in a Hop1-poor region, resolution is mostly dependent on structure specific nucleases (Mus81-Mms4, Yen1, Slx4-Slx1). This pattern is altered in *pch2* mutants where Hop1 appears to be more homogenously distributed.

The experimental strategy, experiments and interpretations are very clear and convincing. These results provide significant, interesting and novel insights into the regulation of meiotic recombination by properties of chromosome structure.

Major comment:

The only experimental validation missing in this paper is to demonstrate the requirement for *ssn* in *pch2* mutant and thus to analyze intermediates, COs and NCOs in *pch2 mms4 yen1 slx1* mutant.

General comments:

1) Several observations and interpretations should be more precisely discussed:

How these observations relate to previous genetic studies on the effect of MutLg and *ssn* on CO in different intervals?

2) How do the authors explain that the ratio NCO/CO (for VDE induced events) is similar is both contexts (His4 vs. Ura3)? According to the model proposed, higher density of meiosis axial elements should lead to increase capture of ZMM and decreased NCO/CO.

3) Why JMs are detected in *ssn* mutant and not or weakly in Mlh3 mutant? In addition, if intermediates are distinct (nicked single/double HJ?) how could resolvases partially substitute, how could Mlh be used as a backup? The discussion in the first paragraph of the subsection “The interplay of resolvase activities is chromosome context-dependent”, does not clarify these questions.

In the absence of Spo11: CO are Mlh-independent, how is Hop1 or Hop1 phosphorylation distributed?

*Reviewer #2:*

Much information has been accumulated concerning the genetic control of meiotic recombination pathways that lead to formation of crossover or noncrossover recombination products. However, much of this work tends to focus on one or two individual genomic loci, so it has remained unclear to what extent observed genetic pathways operate uniformly (or not) across the genome. This intriguing paper documents a substantial difference between two positions in the yeast genome for use of MutLgamma and other Holliday junction resolving activities. The experimental design is clever, taking advantage of site-specific DSBs made by the VDE nuclease. The data are of excellent quality and the striking differences between insertions at HIS4 and URA3 are clearly documented. This is an interesting and important study.

There were a few concerns about the framing of some of the interpretations and about how some of the data are displayed. While these are important points, they can be easily remedied with text fixes.

1) A significant concern has to do with interpreting differences between the HIS4 and URA3 loci in terms of chromosome structure and/or abundance of Hop1 or other chromosome axis proteins. The main issue is that the authors are trying to argue based on a two-point correlation, but there are a large number of possible differences between these loci aside from the ones the authors focus on, and it remains possible that the direction of the correlation seen with these two loci remains purely coincidental. The *pch2* experiments are consistent with the authors' preferred interpretation, but fall short of providing convincing support: while it is true that Hop1 occupancy changes, it is also the case that other things probably change as well in the absence of Pch2. So, it is a stretch to ascribe the effects of this mutation as being solely or even primarily via effects on Hop1 localization. What this paper clearly demonstrates is context dependence for recombination protein utilization. The title of the paper is thus spot-on. But the Discussion needs to be more cautious in interpreting the possible causes of the context dependence, because there is just not enough information here to make strong conclusions.

2) In the subsection “Local chromosome structure influences meiotic CO formation”, second paragraph: There is little or no mention of prior studies of context-dependence for crossover-noncrossover likelihood (Mancera et al. 2008; Serrentino et al. 2013; deBoer et al. 2015) or interhomolog vs. intersister partner choice (Hyppa & Smith 2010; Fowler et al. 2014). The Serrentino paper is mentioned briefly in passing in the preceding paragraph, but I doubt a reader would realize from this mention that paper had documented differences in crossover vs. noncrossover outcome between different loci. There are also studies documenting the different recombination behavior for DSBs within pericentromeric regions (Chen et al. 2008; Vincenten et al. 2015). These prior studies should be discussed in comparison to the context dependence documented here. (This is also relevant to the statement in the subsection “Concluding remarks”, at the end of the first paragraph.)

3) Throughout: SEM is used for error bars, but in most cases this is for experiments with only two replicates. SEM is rarely a good choice for displaying information about experimental variability, but is particularly fraught (as is SD) with just two measurements. It would be better to use error bars to indicate range when there are only two measurements, and SD (or range) for instances where there are larger numbers of replicates. A related point: in Figure 2—figure supplement 1, it's a little unclear if the error bars are SEM here, but if so, the data seem to indicate >100% recovery of arg4-VRS103 for at least one of the experiments at URA3. Is this right? In any case, it would be better to use univariate scatterplots to show the individual values from each replicate rather than using bar graphs with SEM error bars (or more complicated error bars as in Figure 3. If the authors wish to stick to bar graphs, then the error bars should show the range. This paper in PLoS Biology provides an excellent discussion of pitfalls for bar graphs and suggests other strategies for data display: http://journals.plos.org/plosbiology/article?id=10.1371/journal.pbio.1002128

*Reviewer #3:*

In this study Medhi et al. have analyzed the contribution of different Holliday junction resolvases to VDE-initiated CO-formation in recombination reporters inserted at HIS4 and URA3. HIS4 is "hot" whereas URA3 is "cold" for occupancy by the chromosome axis proteins Hop1 and Red1.

The authors find that VDE-initiated COs at HIS4 are strongly dependent on MutLγ, similar to those initiated by Spo11. VDE-induced COs at URA3 are more dependent on SSNs. The authors also show that *pch2∆* mutants, which have smaller differences in Hop1 occupancy between HIS4 and URA3, display smaller locus-dependent differences in formation of COs by MutLγ. The authors conclude that local chromosome context is important for the biochemistry of CO formation during meiosis.

The data is of high quality and the work is overall interesting. If the authors address the issues listed below, I would strongly recommend it for publication in *eLife*.

1) The authors infer general chromosome behavior from 2 loci. It would be important to expand the analysis to at least another pair of "hot" and "cold" loci. Alternatively, the authors could compare published ChIP-seq data of Hop1 with available genome wide recombination maps from resolvase mutants.

2) Since VDE cuts both sister chromatids, could the authors reduce VDE expression to obtain a more comparable type of cleavage to Spo11. Otherwise, it may be inappropriate to extrapolate the obtained results to those of naturally occurring DSBs.

3) In Figure 3, the authors show that CO formation at HIS4 is reduced by 60% in *mlh3* mutants, 30% in *ssn* mutants and 75% in a combination of the two. How are COs made in such mutant and why isn't the reduction at least the sum of both *mlh3* and *ssn* mutants (expected ~90% reduction)? The exact opposite is observed at URA3: The *mlh3 sse* mutant is more defective than the sum of the two pathways.

4) In the subsection “Altered Hop1 occupancy in *pch2* mutants is associated with altered MutLγ− dependence of VDE-initiated Cos”, last paragraph: Why would COs that are Mlh3-dependent increase at URA3 relative to HIS4 (37% vs. 20%) in *pch2∆* mutants. If the Hop1 occupancy is similar, as argued by the authors, why is this value so different?

5) The finding that MLH3-dependent COs are absent in spo11 mutants is not surprising. However, it is surprising that the formed COs are also largely SSN-independent (~70%). How do the authors explain this? The authors should analyze the *mlh3∆ ssn∆* mutant. It is possible that SSNs can backup completely for MLH3 in this case (and MLH3 provides a significant backup for SSNs). Which would suggest that actually Mlh1-Mlh3 does at least part of the job in the WT, but one does not see it because SSNs can compensate for it.

6) How do the authors explain that the dependency on pathway usage at "hot" and "cold" regions is so limited. 60% vs. 30% contribution is not what one would expect if the genome is really partitioned and this is a major biochemical determinant of pathway usage. The authors should discuss this in more detail.

7) CO and NCO frequency is the same at URA3 and HIS4 (Figure 2). If at one locus HJ cleavage is mediated by MutLγ and at the other by SSNs, wouldn't one expect a different NCO/CO ratio? Nucleases have been shown by the Lichten lab (Dayani 2011) to generate a mixture of both, while MutLγ would generate exclusively COs (Zakharyevich 2012).

---

## [Author Response]

[…]

*Essential revisions:*

*In order to fully validate the interpretations, some additional experiments are needed using the tools that the authors have used in the present experiments. In addition, the authors should not over interpret the data which is based on the comparison of two loci (HIS4 and URA3) with a correlation with HOP1 occupancy. There are a large number of possible differences between these loci aside from the ones the authors focus on, and it remains possible that the direction of the correlation seen with these two loci remains purely coincidental. Pch2 mutant fits the expectation but absence of Pch2 has likely several other consequences. The Abstract and conclusions should be modified accordingly. Use of SEM should me revised.*

Please see below for additional work and manuscript changes that address these concerns. We have rewritten the Abstract to modify its overall emphasis, and have also added text to the Discussion that explicitly addresses the above concern about coincidence:

“The observation that some COs at *HIS4* are SSN-dependent, even though most are MutLγ-dependent (Figure 3), indicates that this division is not absolute. […] It remains possible that the association seen at *HIS4* and *URA3*, rather than being directly causative, reflects another underlying aspect of meiotic chromosome structure or function, and that other differences betweenthese two loci cause the observed differences in resolvase usage.”

It should be noted that an emerging consensus is that Pch2’s primary activity involves remodeling HORA-domain proteins (see Rosenberg and Corbett, JCB 2015 for discussion). This makes it likely that the varied meiotic phenotypes of *pch2* mutants are all a consequence of altered Hop1 distributions, but of course altering Hop1 occupancy will affect many different meiotic processes; thus, the text above, in particular the last two sentences.

*1) In order to evaluate the generality of their findings (see general comment above), the authors could compare published ChIP-seq data of Hop1 with available genome wide recombination maps from resolvase mutants.*

This is a great idea. Unfortunately, currently there are not sufficient tetrad data to make a locus-by-locus comparison. The only available SSN mutant data are for a *mms4* meiotic depletion strain (Oke et al., PLOS Genetics 2014), with a total of 596 crossovers in 7 tetrads analyzed, or a scored crossover density of about 1/17kb. Nishant K.T. and collaborators have (unpublished) crossover data for 19 *mlh3* tetrads from an S288c-YJM789 hybrid strain that they have made available to us, but even at this higher crossover number (1224, 1CO/8 kb), there are not enough to confidently score differences between wild-type and *mlh3* on a locus-by-locus basis. We are currently exploring strategies to divide the genome into bins with different Hop1 enrichment levels and examine relative Mlh3-dependence of crossovers in each bin, but this is a complex problem that is going to require considerable work before we even know if the current *mlh3* dataset is of sufficient size.

*2) One important experiment missing in this paper is to demonstrate the requirement for ssn in pch2 mutant and thus to analyze intermediates, COs and NCOs in pch2 mms4 yen1 slx1 mutant.*

We did these experiments and they are presented in Figure 4 and Figure 4—figure supplement 1 and Figure 4—figure supplement 2; corresponding text has also been changed (subsection “Altered Hop1 occupancy in *pch2* mutants is associated with altered MutLγ−dependence of VDE-initiated Cos”, last paragraph).

3) Since there are only two replicate datasets for several analyses, error bars should show range rather than SEM for the time-course plots. Bar graphs should be replaced with univariate scatter plots, but if the authors wish to retain the bar graphs, then error bars should show range, not SEM. This paper in PLoS Biology provides an excellent discussion of pitfalls for bar graphs and suggests other strategies for data display: http://journals.plos.org/plosbiology/article?id=10.1371/journal.pbio.1002128

Error bars were changed to range in all but Figure 3 and Figure 4, where error bars were removed (see below). Figure legends were appropriately changed.

We did not convert bar graphs to scatter plots, as they do not report primary data. Primary data are presented as line graphs and also in [Supplementary-material SD2-data]. The bar graphs are used to summarize features of the data, with the goal of visually communicating conclusions.

Weissgerber et al., cited in the reviewers’ comments, object to bar (and line) graphs because they do not “allow readers to critically evaluate continuous data”. In our paper, data are clearly presented in other figure panels and in a data supplement, so the interested reader has plenty of opportunity for evaluation. Weissgerber et al. dislike bar graphs because they conceal differences in distributions (including outliers), in sample size, and in relationships between dependent variables. In our data, there are no differences in sample size (2, in each case), in distribution relative to the mean (can’t be, with 2 values), and variables are independent. Therefore, the issues that motivate Weissgerber et al. are not relevant to our paper.

In the case of Figure 3 and Figure 4, panels C and D, it is not possible to convert error bars to “range”. This is because the values are the mean of 8 and 9 hr samples in two independent experiments with the indicated mutant (all 4 values averaged), divided by a similar mean for the indicated wild-type strain. Since the values are the ratio of two means, range is not applicable. In addition, values are a mix of dependent (8 and 9 hr samples from the same time course) and independent values (everything else), so formally it is not legitimate to calculate standard deviations. Instead, we removed error bars and representations of significance from these panels, and figure legends have been appropriately adjusted. We believe that these bar graphs still have value (see below), in that they enhance comprehensibility. We would prefer to retain them, but will remove them if requested. If error bars are deemed necessary, then we can calculate standard deviations for these ratios (we agree that S.E.M. was not correct), keeping in mind that such a calculation is not strictly legitimate.

(The following contains Michael Lichten’s views, not necessarily those of the other authors, and is included here by way of discussion with the editor, editorial staff, and reviewers. Please feel free to delete it from the public review record or include it, as you see fit.)

Despite current trends to the contrary, there is a definite value to summary plots and statistics, if they enhance clarity and comprehensibility but are not the only form in which data are presented, and if they are not used to hide data features that are relevant to the analysis. Univariate scatter plots (a.k.a. “confetti plots”), currently so popular, actually can make data less comprehensible and more obscure, especially when sample sizes are so large that individual points cannot be distinguished. It is hoped that the suggestions of Weissgerber et al., which are certainly well taken, will not be blindly imposed on every paper that is submitted to *eLife*, but rather will be used in situations where they are appropriate.

*4) The authors should explicitly indicate that in most meiosis VDE cuts both sister chromatids and that the consequences of this on pathway choices are unknown.*

The following text was added:

“Thus, in most cells, both sister chromatids are cut by VDE (Gimble and Thorner, 1992; Neale et al., 2002). In contrast, Spo11-DSBs infrequently occur at the same place on both sister chromatids (Zhang et al., 2011). While the consequences of this difference remain to be determined, we note that inserts at both *HIS4* and *URA3* are cleaved by VDE with equal frequency (Figure 2). Thus, any effects due simultaneous sister chromatid-cutting should be equal at the two loci.”

*5) In the subsection “Local chromosome structure influences meiotic CO formation”, second paragraph: There is little or no mention of prior studies of context-dependence for crossover-noncrossover likelihood (Mancera et al. 2008; Serrentino et al. 2013; deBoer et al. 2015) or interhomolog vs. intersister partner choice (Hyppa & Smith 2010; Fowler et al. 2014). The Serrentino paper is mentioned briefly in passing in the preceding paragraph, but I doubt a reader would realize from this mention that paper had documented differences in crossover vs. noncrossover outcome between different loci. There are also studies documenting the different recombination behavior for DSBs within pericentromeric regions (Chen et al. 2008; Vincenten et al. 2015). These prior studies should be discussed in comparison to the context dependence documented here. (This is also relevant to the statement in the subsection “Concluding remarks”, at the end of the first paragraph.)*

We agree that previous findings were given short shrift, and included more of this information in the paper. However:

1) Mancera et al. say that CO/NCO ratios differ at different loci, but this is likely a consequence of small sample size and uneven distribution of polymorphic markers.

2) Serentino et al. showed that three DSB sites with lower Zip3 occupancy/DSB ratios had fewer COs (measured by genetic distance, cM) than a DSB site with higher Zip3 occupancy. Please note that, because NCOs were not scored, this could be because of changes in CO/NCO *or* in IH/IS ratios. Their calculations used ssDNA data from Buhler et al. as a proxy for DSB levels; when the calculation is made using Pan et al.’s more recent Spo11-oligo data, the differences between loci become much less marked, and Zip3 occupancy correlates fairly well with Spo11 oligo levels (ML, unpublished).

Rather than include an extensive discussion of these issues, we wrote the following:

“Serrentino et al. (2013) showed that enrichment for the budding yeast ZMM protein, Zip3, at DSB sites is correlated with interhomolog CO levels. […] Locus-specific differences in CO/NCO ratios also have been observed in mouse meiosis (de Boer et al., 2015), locus-specific differences in partner choice have been reported in *S. pombe* (Hyppa and G. R. Smith, 2010), and crossover suppression by centromeres is observed in many species (Talbert and Henikoff, 2010).”